# CLEME: Debiasing Multi-reference Evaluation for Grammatical Error Correction

**Jingheng Ye[1*], Yinghui Li[1*], Qingyu Zhou[2], Yangning Li[1,3],**
**Shirong Ma[1], Hai-Tao Zheng[1,3†], Ying Shen[4]**

[1]Tsinghua Shenzhen International Graduate School, Tsinghua University
[2]OPPO Research Institute, [3]Peng Cheng Laboratory
[4]School of Intelligent Systems Engineering, Sun-Yat Sen University
{yejh22,liyinghu20}@mails.tsinghua.edu.cn

## Abstract

Evaluating the performance of Grammatical Error Correction (GEC) systems is a challenging task due to its subjectivity. Designing an evaluation metric that is as objective as possible is crucial to the development of GEC task. However, mainstream evaluation metrics, i.e., reference-based metrics, introduce bias into the multi-reference evaluation by extracting edits without considering the presence of multiple references. To overcome this issue, we propose **C**hunk-**LE**vel **M**ulti-reference **E**valuation (**CLEME**), designed to evaluate GEC systems in the multi-reference evaluation setting. CLEME builds chunk sequences with consistent boundaries for the source, the hypothesis and references, thus eliminating the bias caused by inconsistent edit boundaries. Furthermore, we observe the consistent boundary could also act as the boundary of grammatical errors, based on which the $F_{0.5}$ score is then computed following the correction independence assumption. We conduct experiments on six English reference sets based on the CoNLL-2014 shared task. Extensive experiments and detailed analyses demonstrate the correctness of our discovery and the effectiveness of CLEME. Further analysis reveals that CLEME is robust to evaluate GEC systems across reference sets with varying numbers of references and annotation styles [1].

## 1 Introduction

Grammatical Error Correction (GEC) is a task that involves making *local substitutions* to correct grammatical errors in a given ungrammatical text (Bryant et al., 2022; Ma et al., 2022; Ye et al., 2022; Ma et al., 2023). The practical value of GEC in daily life has led to increasing attention being paid to this task (Li et al., 2021, 2022a,b; Kaneko

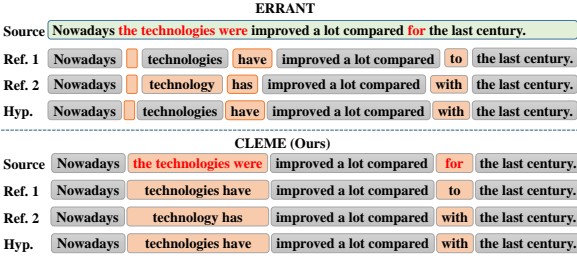

Figure 1: A comparison of edits automatically extracted by ERRANT and **CLEME**. An orange block is an edit.

et al., 2022; Li et al., 2023; Ye et al., 2023a,b; Zhang et al., 2023). However, it is intractable to evaluate GEC systems due to the highly subjective nature of the task and the low inter-annotator agreement (IAA) (Choshen and Abend, 2018). Therefore, most datasets improve compatibility by incorporating multiple references to guarantee a more realistic evaluation of the model performance.

There are two broad categories of GEC metrics: reference-based and reference-less. Reference-based metrics evaluate GEC systems by comparing their hypotheses and human-annotated references in terms of edits (Dahlmeier and Ng, 2012; Bryant et al., 2017) or n-grams (Napoles et al., 2015). Reference-less metrics are proposed to evaluate GEC systems without references. However, Deutsch et al. (2022) demonstrate that reference-less metrics are inherently biased and limited in their ability to evaluate generated text. Therefore, we focus on reference-based metrics, which can evaluate in an interpretable manner, thus providing useful insights for model analysis.

Figure 1 illustrates how existing reference-based metrics, such as ERRANT, extract the edit and then compute the $F_{0.5}$ score by comparing hypotheses and references. However, these metrics often fail to consider multiple references, which can result in bias during multi-reference evaluation. We argue that this bias arises because the current approach rewards equally good corrections unfairly. For in-

---

[*]indicates equal contribution.
[†]Corresponding author: Hai-Tao Zheng. (E-mail: zheng.haitao@sz.tsinghua.edu.cn)
[1]All the source codes of CLEME are released at https://github.com/THUKElab/CLEME.

stance, the ungrammatical phrase *the technologies were* is equally well-corrected by both Ref. 1 and Ref. 2. However, if a hypothesis aligns with Ref. 1's corrections (i.e., [*the* → $\epsilon$] and [*were* → *have*], TP=2), it will be rewarded less than the corrections of Ref. 2 (i.e., [*the* → $\epsilon$], [*technologies* → *technology*] and [*were* → *has*], TP=3).

In this paper, we propose **C**hunk-**LE**vel **M**ulti-reference **E**valuation (**CLEME**), which enables unbiased $F_{0.5}$ scores for GEC multi-reference evaluation. Inspired by (Gotou et al., 2020), CLEME transforms the source, the hypothesis and all the references into chunk sequences with consistent boundaries, thereby eliminating the bias in GEC multi-reference evaluation.

Existing metrics assume that corrections of grammatical errors are dependent. That is, whenever there is more than one reference for a source, the metrics try each reference in turn, and then the highest score is taken as the final score. However, we observe that grammatical errors corrections in terms of chunks can be considered **approximately independent**. For example, the ungrammatical phrases *the technologies were* and *for* shown in Figure 1 can be corrected independently, i.e., the correction of *the technologies were* has no bearing on the correction of *for*. Based on this observation, we compute $F_{0.5}$ scores following the assumption that corrections of grammatical errors are independent. Specifically, we iterate through the chunks of a hypothesis and consider a chunk correct if it matches any of the corresponding chunks in the references. In this case, the hypothesis in Figure 1 would be rewarded 2TP, rather than 1TP and 1FP, which is the traditional case. To demonstrate the effectiveness and robustness of CLEME, we conduct experiments on six English reference sets with varying numbers of references and annotation styles, either calculating the $F_{0.5}$ score at the corpus- or sentence-level.

In summary, our contributions are three folds:

(1) We propose CLEME, a reference-based metric that evaluates GEC systems at the chunk-level, aiming to provide unbiased $F_{0.5}$ scores for GEC multi-reference evaluation.

(2) We observe that the corrections of grammatical errors in terms of chunks are approximately independent. Therefore, we propose to compute $F_{0.5}$ scores based on the correction independence assumption.

(3) Extensive experiments and human evaluation are conducted to confirm the effectiveness and robustness of our approach.

## 2 Preliminary Study

### 2.1 Consistent Boundaries

We determine consistent chunk-level boundaries by chunk partition process to debias the multi-reference evaluation, as depicted in Figure 2. We first extract the edit sets of the hypothesis and references, and then merge the overlapping edits into a chunk. It's worth noting that the source, hypothesis and references are all segmented into *chunk sequences* with the same number of chunks, regardless of the number of their tokens. This process is straightforward since we can locate and examine all possible corrections of an erroneous chunk. For example, the chunk *by the* can be corrected in two ways, i.e., *with* in Ref. 1 and *through* in Ref. 2. The resulting chunks fall into three categories: 1) **unchanged chunks**, which contain the same text segments as the source sentence, 2) **corrected chunks**, which consist of non-empty text segments different from the source sentence, and 3) **dummy chunks** are empty chunks.

### 2.2 Boundaries of Grammatical Errors

Figure 2 illustrates the merging of overlapping edits into either corrected or dummy chunks, which are then separated by unchanged chunks. This raises the question, *are chunk boundaries the boundaries of grammatical errors?*

**Dataset.** To answer the question, we conduct experiments on BN-10GEC (Bryant and Ng, 2015). The dataset comprises 1,312 source sentences that are identical to the CoNLL-2014 test data (Ng et al., 2014). Each source sentence is associated with 10 references annotated by 10 native English speakers, including two official annotators of CoNLL-2014, the first author of the paper, and seven freelancers recruited via an online recruitment website.

**Experiment Setup.** For each source sentence, we sample 9 references and run the chunk partition process described in Section 2.1. The resulting chunk sequences are determined collectively by all 9 references. The edits of the remaining reference $\{e_1, \cdots, e_M\}$ are then used to calculate the following three statistics: 1) The In-Corrected-Chunk (ICC) ratio indicates the proportion of edits included by corrected/dummy chunks of the other

references. An edit is included by a chunk if the interval of the edit falls within that of the chunk. 2) The In-Unchanged-Chunk (IUC) ratio gives the proportion of edits included by unchanged chunks of the other references. 3) The Cross-Chunk (CC) ratio computes the proportion of edits that extend beyond the original boundaries. These statistics are calculated as follows:

$$\text{ICC} = \frac{1}{M} \sum_{i=1}^{M} f_1(e_i), \qquad (1)$$

$$\text{IUC} = \frac{1}{M} \sum_{i=1}^{M} f_2(e_i), \qquad (2)$$

$$\text{CC} = 1 - \text{ICC} - \text{IUC}, \qquad (3)$$

where $M$ is the number of edits from the remaining reference. If the edit $e_i$ is included in a corrected/dummy chunk, the function $f_1(e_i)$ returns 1, otherwise 0. Likewise, if the edit $e_i$ is included in an unchanged chunk, the function $f_2(e_i)$ returns 1, otherwise 0. We sample 9 different references for chunk partition in each run and repeatedly calculate the statistics using the remaining reference.

**Results.** As shown in Table 1, the number of corrected and dummy chunks are less than that of edits since overlapping edits are merged into a chunk. A total of 90.66% edits are included by the corrected/dummy chunks, which suggests the grammatical errors to be corrected have been considered by the other references. However, only 7.74% edits are included by corrected chunks, indicating that these edits may be over-corrected since the other references believe no grammatical errors needed correction. Interestingly, 1.61% edits cross the chunk boundaries, suggesting that the chunk boundaries are stable enough to serve as the boundaries of grammatical errors to some extent. Additionally, human evaluation in Section 4.2 could be used as another argument to support this conclusion. Therefore, we have the following assumption.

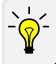 Correction independence assumption: grammatical error corrections are independent.

That is, the correction of a grammatical error does not impact the correction of other grammatical errors. With this assumption, $F_{0.5}$ scores can be calculated using an alternate method, which will be introduced in Section 3.

| Item | Number (perc.) | Length |
|------|----------------|--------|
| Sentences | 1,312 | 23.0 |
| References | 13,120 | 22.9 |
| Edits | 36,677 | 1.0 |
| Unchanged Chunks | 93,469 (77.63%) | 2.5 |
| Corrected/Dummy Chunks | 26,948 (22.37%) | 2.4 |
| ICC | 33,251 (90.66%) | - |
| IUC | 2,837 (7.74%) | - |
| CC | 589 (1.61%) | - |

Table 1: Statistics of the BN-10GEC dataset.

## 3 Method

### 3.1 Chunk Evaluation

As shown in Figure 2, each chunk consists of edit operation(s), start index, end index, and correct tokens. Conventional reference-based metrics such as MaxMatch ($M^2$) and ERRANT compute $F_{0.5}$ scores based on the correction dependence assumption. They evaluate the performance for each reference separately and select the one that yields the best result for the source sentence. **CLEME-dependent** also computes $F_{0.5}$ scores in this way by treating corrected/dummy chunks as edits. On the other hand, **CLEME-independent** is proposed to compute $F_{0.5}$ scores based on the correction independence assumption. A corrected/dummy chunk from the hypothesis is considered correct if it matches one of the corresponding chunks from the references. It is worth noting that CLEME is able to fully inherit pre-classified errors from ERRANT, where each corrected/dummy chunk may consist of multiple error with different types.

### 3.2 Length Weighting

The average length of chunks is much longer than that of edits shown in Table 1, resulting in the unfairness of chunk evaluation if a longer chunk is rewarded equally with a shorter one. Therefore, we add length weighting to the chunk evaluation. The intuition of length weighting is to compensate for long chunk matching. The weights of True Positives (TPs), False Positives (FPs), and False Negatives (FNs) are computed as follows:[2]

$$w^{\text{TP}} = \text{clip}\left(\frac{\alpha_1}{1 + (\alpha_1 - 1)\exp(\ell - x)}, c_{\min}, c_{\max}\right), \quad (4)$$

$$w^{\text{FP}} = \text{clip}\left(\frac{\alpha_2}{1 + (\alpha_2 - 1)\exp(x - \ell)}, c_{\min}, c_{\max}\right), \quad (5)$$

---

[2]We do not apply length weighting to TNs since it is unnecessary for $F_{0.5}$ scores.

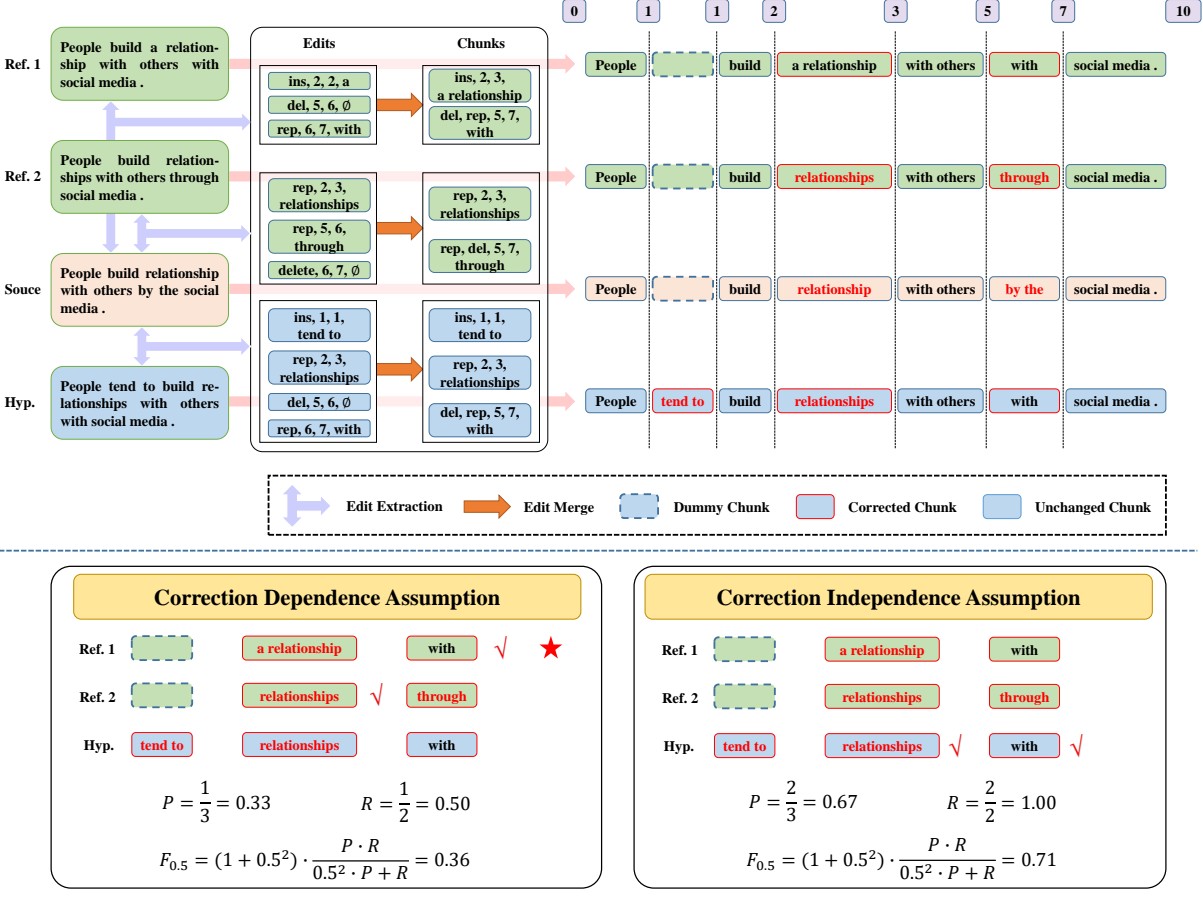

Figure 2: Overview of our approach CLEME. CLEME first 1) extracts edits of the hypothesis and the references, 2) merges the overlapping edits into chunks, and then 3) computes the $F_{0.5}$ scores based on two different assumptions.

$$w^{\text{FN}} = \text{clip}\left(\frac{\alpha_3}{1 + (\alpha_3 - 1)\exp(\ell - x)}, c_{\min}, c_{\max}\right), \quad (6)$$

where $\alpha_1$, $\alpha_2$ and $\alpha_3$ are scale factors for TPs, FPs and FNs respectively, $x$ is the length of the chunk, $\ell$ is the average length of chunks, and the function $\text{clip}(v, a, b)$ clips the value $v$ between $a$ and $b$. The curves of length weighting are depicted in Figure 3. Formally, given a system corrected/dummy chunk set $C^H$ and a gold corrected/dummy chunk set $C^R$, we apply length weighting on each chunk to compute precision, recall and $F_{0.5}$ as follows:

$$P = \frac{\sum\limits_{c \in C^H \cap C^R} w_c^{\text{TP}}}{\sum\limits_{c \in C^H \cap C^R} w_c^{\text{TP}} + \sum\limits_{c \in C^H \setminus C^R} w_c^{\text{FP}}}, \quad (7)$$

$$R = \frac{\sum\limits_{c \in C^H \cap C^R} w_c^{\text{TP}}}{\sum\limits_{c \in C^H \cap C^R} w_c^{\text{TP}} + \sum\limits_{c \in C^R \setminus C^H} w_c^{\text{FN}}}, \quad (8)$$

$$F_\beta = (1 + \beta^2) \cdot \frac{P \cdot R}{(\beta^2 \cdot P) + R}, \quad (9)$$

where $\beta = 0.5$ is usually used, which weighs precision twice as much as recall.

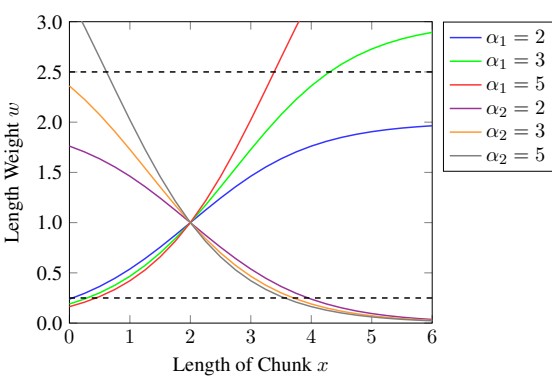

Figure 3: Curves of length weighting with different $\alpha$ for $\ell = 2$. All the curves pass through the point $(\ell, 1.0)$. A curve with a larger scale factor has a greater slope.

### 3.3 Corpus-level v.s. Sentence-level

We compute $F_{0.5}$ scores of GEC systems at both corpus-level and sentence-level following (Gong et al., 2022). Corpus-level metrics compute an $F_{0.5}$ score over the entire dataset. Sentence-level metrics compute an $F_{0.5}$ score over each sentence of the dataset and evaluate GEC systems by us-

ing the average $F_{0.5}$ score. CLEME-dependent and CLEME-independent are corpus-level metrics, and their sentence-level variants are respectively **SentCLEME-dependent** and **SentCLEME-independent**. Both levels of the GEC metric are developed to provide more user-friendly options. Sentence-level metrics should be used if consistent evaluation weight for each sample is desired. This ensures that the evaluation result of each sample has the same influence on the final score. On the other hand, if harder samples containing more edits should have larger weight, then corpus-level metrics should be used instead.

## 4 Experiments

### 4.1 Correlations with Human Judgments

**Dataset.** To verify the effectiveness of CLEME, we measure correlations between reference-based metrics and human judgments on multiple English reference sets, including CoNLL-2014 (Grundkiewicz et al., 2015), BN-10GEC (Bryant and Ng, 2015) and SN-8GEC (Sakaguchi et al., 2016). All the reference sets are based on CoNLL-2014 (Ng et al., 2014), consisting of 1,312 source sentences. SN-8GEC collected 8 references sets of annotations from both experts and non-experts, including 4 sets of minimal edits and 4 sets of fluency edits (2 by experts and 2 by non-experts). Reference sets statistics are reported in Appendix A.

The human judgments for the outputs of 13 GEC systems (including the unchanged source text) are presented by (Grundkiewicz et al., 2015), where eight native speaker were asked to rank the output of all the systems from best to worst. Two system ranking lists are generated using Expected Wins (EW) (Macháček and Bojar, 2013) and TrueSkill (TS) (Sakaguchi et al., 2014) respectively.

**Experiment Settings.** Following (Gong et al., 2022; Chollampatt and Ng, 2018), we compute the Pearson $\gamma$ and Spearman correlation coefficient $\rho$ between reference-based metrics and human judgments based on corpus-level ranking. We tune the hyperparameters on CoNLL-2014 and keep the hyperparameters on the other reference sets, in order to demonstrate the adaptability of our approach. The detailed hyperparameters of CLEME are reported in Appendix B.

**Evaluation Metrics.** We compare our approach with the following reference-based metrics, includ-

ing corpus- and sentence-level variants [3]:

- **GLEU** and **SentGLEU** (Napoles et al., 2015) are n-gram based metrics, which reward hypothesis n-grams that overlap with the reference but not the source and penalize hypothesis n-grams that overlap with the source but not the reference.

- **$M^2$** and **SentM$^2$** (Dahlmeier and Ng, 2012) dynamically extract the hypothesis edits with the maximum overlap of gold annotations.

- **ERRANAT** and **SentERRANT** (Bryant et al., 2017) extract edits by utilizing a linguistically-enhanced alignment algorithm.

- **PT-$M^2$** and **SentPT-M$^2$** (Gong et al., 2022) are recently proposed reference and PLM-based GEC metric, which score edits using the knowledge of pre-trained language model.

Additionally, CLEME can evaluate GEC systems by accuracy scores, which is usually not implemented by conventional reference-based metrics. Please refer to Appendix C for the introduction and analyses of evaluating GEC systems by accuracy.

**Results.** Table 2 reports the correlations between reference-based metrics and human judgments. For the corpus-level metrics, GLEU achieves the highest correlations on BN-10GEC and NE-fluency reference sets. However, GLEU suffers from negative correlations on NE-Minimal, which is caused by low-quality annotations [4] of NE-Minimal, indicating that GLEU may not be a robust metric, consistent with the findings of (Sakaguchi et al., 2016). ERRANT performs slightly better than $M^2$ on most reference sets, while PT-$M^2$ is a strong corpus-level metric, which achieves the highest or comparable correlations on all reference sets at the cost of more than $10\times$ running time than other reference-based metrics. Our proposed CLEME-dependent and CLEME-independent make better use of consistent chunk boundaries, thus performing slightly better than $M^2$ and ERRANT on most reference sets. Notably, CLEME-independent achieves comparable performance to CLEME-dependent, showing the

---

[3] We do not experiment with I-measure (Felice and Briscoe, 2015) due to its negative correlation and high computing complexity (Grundkiewicz et al., 2015).

[4] The phenomenon exists on all sentence-level metrics. We remove unchanged references from some reference sets to avoid it.

| Metric | | CoNLL-2014 | | BN-10GEC | | E-Minimal | | E-Fluency | | NE-Minimal | | NE-Fluency | |
|---|---|---|---|---|---|---|---|---|---|---|---|---|---|
| | | EW | TS | EW | TS | EW | TS | EW | TS | EW | TS | EW | TS |
| $M^2$ | $\gamma$ | 0.623 | 0.672 | 0.547 | 0.610 | 0.597 | 0.650 | 0.590 | 0.659 | 0.575 | 0.634 | 0.582 | 0.649 |
| | $\rho$ | 0.687 | 0.720 | 0.648 | 0.692 | 0.654 | 0.703 | 0.654 | 0.709 | 0.577 | 0.648 | 0.648 | 0.703 |
| GLEU | $\gamma$ | **0.701** | **0.750** | **0.678** | **0.761** | 0.533 | 0.513 | **0.693** | **0.771** | -0.044 | -0.113 | **0.674** | **0.767** |
| | $\rho$ | 0.467 | 0.555 | **0.754** | 0.806 | 0.577 | 0.511 | 0.710 | 0.757 | -0.005 | -0.055 | 0.725 | **0.819** |
| ERRANT | $\gamma$ | 0.642 | 0.688 | 0.586 | 0.644 | 0.578 | 0.631 | 0.594 | 0.663 | 0.585 | 0.637 | 0.597 | 0.659 |
| | $\rho$ | 0.659 | 0.698 | 0.637 | 0.698 | 0.742 | 0.786 | 0.720 | 0.775 | 0.747 | 0.797 | 0.753 | 0.797 |
| PT-$M^2$ | $\gamma$ | 0.693 | 0.737 | 0.650 | 0.706 | 0.626 | 0.667 | 0.621 | 0.681 | 0.630 | 0.675 | 0.620 | 0.682 |
| | $\rho$ | **0.758** | **0.769** | 0.690 | **0.824** | 0.709 | 0.736 | **0.758** | **0.802** | 0.736 | 0.758 | **0.758** | 0.802 |
| CLEME-dependent (Ours) | $\gamma$ | 0.648 | 0.691 | 0.602 | 0.656 | 0.594 | 0.644 | 0.589 | 0.654 | 0.595 | 0.643 | 0.612 | 0.673 |
| | $\rho$ | 0.709 | 0.742 | 0.692 | 0.747 | **0.797** | **0.813** | 0.714 | 0.775 | 0.786 | 0.835 | 0.720 | 0.791 |
| CLEME-independent (Ours) | $\gamma$ | 0.649 | 0.691 | 0.609 | 0.659 | 0.593 | 0.643 | 0.587 | 0.653 | 0.601 | 0.647 | 0.611 | 0.672 |
| | $\rho$ | 0.709 | 0.731 | 0.692 | 0.747 | 0.791 | 0.802 | 0.731 | 0.791 | **0.797** | **0.841** | 0.714 | 0.786 |
| SentM$^2$ | $\gamma$ | 0.871 | 0.864 | 0.567 | 0.646 | 0.805♣ | 0.836♣ | 0.655 | 0.732 | 0.729♣ | 0.785♣ | 0.621 | 0.699 |
| | $\rho$ | 0.731 | 0.758 | 0.593 | 0.648 | 0.806♣ | **0.845♣** | 0.731 | 0.764 | 0.797♣ | 0.846♣ | 0.632 | 0.687 |
| SentGLEU | $\gamma$ | 0.784 | 0.828 | 0.756 | 0.826 | 0.742♣ | 0.773♣ | 0.785 | 0.846 | 0.723♣ | 0.762♣ | 0.778 | 0.848 |
| | $\rho$ | 0.720 | 0.775 | 0.769 | 0.824 | 0.764♣ | 0.797♣ | 0.791 | 0.846 | 0.764♣ | 0.830♣ | 0.768 | 0.846 |
| SentERRANT | $\gamma$ | 0.870 | 0.846 | 0.885 | 0.896 | 0.768♣ | 0.803♣ | 0.806 | 0.732 | 0.710♣ | 0.765♣ | 0.793 | 0.847 |
| | $\rho$ | 0.742 | 0.747 | 0.786 | 0.830 | 0.775♣ | 0.819♣ | 0.813 | 0.764 | 0.780♣ | 0.841♣ | 0.830 | 0.857 |
| SentPT-$M^2$ | $\gamma$ | **0.949** | **0.938** | 0.602♣ | 0.682♣ | 0.831♣ | 0.855♣ | 0.689 | 0.763 | 0.770♣ | 0.822♣ | 0.648 | 0.725 |
| | $\rho$ | **0.907** | **0.874** | 0.626♣ | 0.670♣ | 0.808♣ | 0.819♣ | 0.797 | 0.841 | 0.813♣ | 0.857♣ | 0.742 | 0.786 |
| SentCLEME-dependent (Ours) | $\gamma$ | 0.876 | 0.844 | **0.915** | **0.913** | 0.806♣ | 0.838♣ | 0.849 | 0.886 | 0.742♣ | 0.795♣ | **0.876** | **0.921** |
| | $\rho$ | 0.824 | 0.808 | **0.835** | **0.874** | 0.775♣ | 0.819♣ | 0.824 | 0.863 | 0.797♣ | 0.846♣ | 0.791 | 0.846 |
| SentCLEME-independent (Ours) | $\gamma$ | 0.868 | 0.857 | 0.855♣ | 0.876♣ | 0.821♣ | 0.856♣ | 0.841 | 0.877 | **0.782♣** | **0.831♣** | 0.852 | 0.896 |
| | $\rho$ | 0.725 | 0.758 | 0.659♣ | 0.714♣ | 0.775♣ | 0.819♣ | 0.808 | 0.846 | **0.819♣** | **0.874♣** | 0.762 | 0.825 |

Table 2: Overview of correlations between mainstream GEC metrics and human judgments. We highlight the **highest** score in bold and the second-highest score with underlines. SN-8GEC consists of four reference sets, i.e., E-Minimal, E-Fluency, NE-Minimal and NE-Fluency. ♣ We remove unchanged reference sentences for higher correlations due to low-quality annotations. Otherwise, negative correlations are possible.

effectiveness of computing $F_{0.5}$ scores based on the correction independence assumption.

The majority of the sentence-level metrics outperform their corpus-level counterparts because they weigh samples equally, which is in line with the bias of human annotation. Despite the strong performance of PT-$M^2$, SentPT-$M^2$ achieves lower correlations on BN-10GEC, E-Fluency and NE-Fluency compared to other sentence-level metrics. It suggests that scoring edits using pre-trained language models may not generalize well to unseen reference sets for sentence-level metrics. Our approach aligns better with human judgments than existing reference-based metrics for most reference sets. Specifically, SentCLEME-dependent performs best on BN-10GEC and NE-Fluency, and performs on a par with the best metric on E-Fluency, indicating it is more suitable for fluent reference sets. This phenomenon aligns with our intuition since fluent editing is more likely to follow the correction dependence assumption. In contrast, SentCLEME-independent achieves higher correlations on E-Minimal and NE-Minimal, as we would expect from minimal editing that is more likely to follow the correction independence assumption. These results suggests that reference sets may have a preference towards one of the correction assump-

tions. Additionally, our approach achieves higher correlations on (N)E-Fluency rather than (N)E-Minimal, while SentM$^2$ and SentERRANT perform worse on E-Fluency than E-Minimal. This is because CLEME evaluates GEC systems using longer chunks rather than scrappy edits, which could better reflect whether a grammatical error is fluently corrected. Overall, our approach achieves higher or comparable correlations on sentence-level than existing reference-based methods.

## 4.2 Human Evaluation

Experiments have shown the effectiveness of evaluating GEC systems based on the correction independence assumption. In this section, we aim to demonstrate whether the correction independence assumption makes sense for humans. We define the correction independence of a pair of chunks as the irrelevance of the correction of one chunk to the correction of the other. A simple case is presented in Appendix E. To evaluate this assumption, we conduct human evaluation experiments on 1,000 sentences randomly sampled from BN-10GEC (Bryant and Ng, 2015). Three annotators were asked to judge whether a pair of chunks is correction-independent.

Table 3 reports the ratio of correction indepen-

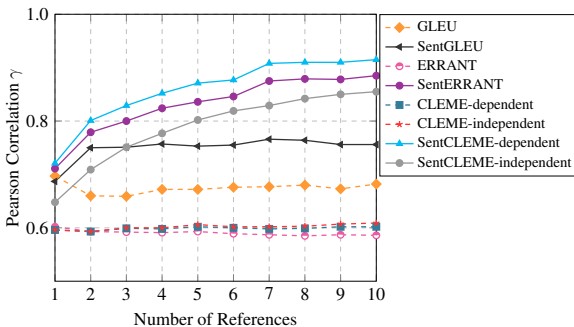

Figure 4: Effect of references number on BN-10GEC.

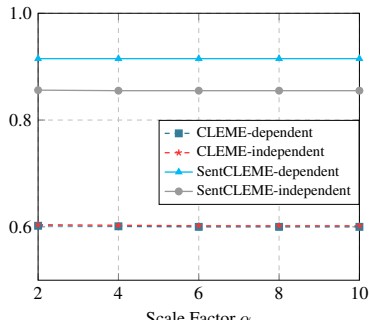

Figure 5: Effect of scale factors on BN-10GEC.

| Annotator | Ratio of Correction Independence |
|---|---|
| $A_1$ | 90.85% |
| $A_2$ | 93.55% |
| $A_3$ | 91.14% |

| Annotator | Cohen's-$\kappa$ |
|---|---|
| $A_1$ v.s. $A_2$ | 38.66% |
| $A_1$ v.s. $A_3$ | 43.10% |
| $A_2$ v.s. $A_3$ | 39.34% |

Table 3: A comparison of correction independence annotations across three annotators.

| | Text | FP | FN |
|---|---|---|---|
| **Source** | It has improved compared for the last century. | | |
| **Hyp.** | It has improved compared between the last century. | | |
| | *ERRANT* | | |
| **Ref. 1** | It has improved compared to the century. | 1 | 1 |
| **Ref. 2** | It has improved compared with the century. | 1 | 1 |
| | *CLEME* | | |
| **Ref. 1** | It has improved compared to the century. | 1 | 0 |
| **Ref. 2** | It has improved compared with the century. | 1 | 0 |

Table 4: Cases of ERRANT and CLEME. ERRANT gives FP=1 and FN=1 since the hypothesis does not match one of the edits of references. CLEME gives only FP=1 since the hypothesis tries to correct the error.

dence and Cohen's-$\kappa$ (Cohen, 1960) inter-annotator agreement (IAA) across the three annotators. Results show that more than 90% pairs of chunks are correction-independent for all the annotators, indicating that it is reasonable to evaluate GEC systems based on the correction independence assumption. Moreover, considering the subjectivity of GEC task, the IAA statistics show that it is relatively easy to judge whether a pair of chunks is correction-independent, compared with the previous study (Bryant and Ng, 2015) [5].

## 5 Analysis

### 5.1 False Negative

We observe that the number of false negatives (FNs) identified by CLEME is significantly lower than that of ERRANT. This difference can be attributed to the distinct definitions used by each system. While ERRANT considers FNs as edits in the reference that do not match those made in the hypothesis, CLEME identifies FNs as corrected/dummy chunks in the reference that do not match the chunks in the hypothesis. We argue the definition of ERRANT is problematic, as it tends to

---

[5] Bryant and Ng (2015) attempted to compute IAA at the sentence level. Three raters were asked simply to decide whether 200 sentences were correct or not. The authors reported IAA of just 0.16, 0.4 and 0.23.

overestimate FN counts in grammatical error correction (GEC) systems, which is evident from the examples presented in Table 4. On the other hand, CLEME's definition also includes true negatives (TNs), making it possible to calculate accuracy.

### 5.2 Ablation Study

We present ablation analyses of our approaches on BN-10GEC - we have similar findings on other reference sets. We report Pearson correlations $\gamma$ using Expected Wins ranking. The trend is similar for Spearson correlations and TrueSkill ranking.

**Number of References.** Since CLEME is designed for multi-reference evaluation, it degrades to conventional reference-based metrics such as $M^2$ and ERRANT when only one reference is available. Here we demonstrate how correlations change against an increasing number of available references. The results reported in Figure 4 indicate that the correlations of corpus-level metrics do not change significantly with the increasing number of available references. However, except for SentGLEU, correlations of sentence-level metric are consistently higher than corpus-level metrics, and steadily increase with more references. Therefore, we recommend evaluating GEC systems using sentence-level metrics rather than corpus-level

| | Chunk 1 | Chunk 2 | Chunk 3 | Chunk 4 | Chunk 5 | Chunk 6 |
|---|---|---|---|---|---|---|
| Source | On the other hand , if there are | ways can | help us to control | or | cure the disease , we can | going . |
| Hyp. | On the other hand , if there are | ways that can | help us to control | and | cure the disease , we can | go . |
| Ref. 1 | On the other hand , if there are | ways that can | help us to control | or | cure the disease , we can | go . |
| Ref. 2 | On the other hand , if there are | things that can | help us to control | and | cure the disease , we can | go . |

| | Chunk 1 | Chunk 2 | Chunk 3 | Chunk 4 | Chunk 5 |
|---|---|---|---|---|---|
| Source | On | | one hand , we do not want this potential danger | causing firghtenning affects in | our lives . |
| Hyp. | On | | one hand , we do not want this potential danger | causing frightening affects in | our lives . |
| Ref. 1 | On | | one hand , we do not want this potential danger | having frightening effects in | our lives . |
| Ref. 2 | On | the | one hand , we do not want this potential danger | to have frightening effects on | our lives . |

| | Chunk 1 | Chunk 2 | Chunk 3 | Chunk 4 |
|---|---|---|---|---|
| Source | Especially for the | young people | without marrige , if he/she is | known to have some genetic risk . |
| Hyp. | Especially for the | young people | without marriage , if the latter is | known to have some genetic risk . |
| Ref. 1 | Especially for unmarried | young people | marrige who are | known to have some genetic risk . |
| Ref. 2 | This is especially the case for | young people | who are unmarried , if he/she is | known to have some genetic risk . |

Table 5: Cases of chunk partition. These tables are automatically generated by CLEME. More cases from multiple datasets and language are provided in Appendix F.

metrics for the multi-reference evaluation setting.

**Parameter Sensitivity Analysis.** The scale factors introduced in Section 3.2 dictate how much the weights of chunks change with their length. We report the corrections for various scale factors, as shown in Figure 5. The results demonstrate that CLEME is resilient to hyperparameter selection.

### 5.3 Case Study

Table 5 presents additional examples of CLEME. In the top group, chunk 2 and chunk 4 of the hypothesis respectively match those of Ref. 1 and Ref. 2. In this case, CLEME-dependent gives TP=1 and FP=1, while CLEME-independent gives TP=2. In the second group, the hypothesis exactly corrects the ungrammatical word *firghtenning* in chunk 4. However, it cannot be rewarded since the entire chunk is not corrected. In the bottom group, two given references have made extensive modifications, with an unchanged chunk *young people*. Evaluating hypotheses in terms of chunks is generally more challenging than fragmented edits, but it provides a more comprehensive diagnosis.

Even though there are larger grammatical errors spanning a significant portion of a sentence, CLEME would not necessarily *collapse*, i.e., producing one single correction chunk spanning the entire sentence. If collapse happens, the quality of the reference set should be checked first. This is because that collapse happens only if the input sentences of chunk partition are completely different, resulting in a trivial chunk partition result, which is an extreme case that has not been observed in our

| Reference-based Metrics | Granularity | Score | Deterministic |
|---|---|---|---|
| M$^2$ (Dahlmeier and Ng, 2012) | Phrase-level Edit | F$_\beta$ | ● |
| GLEU (Napoles et al., 2015) | N-gram | Weighted Precision | ○ |
| ERRANT (Bryant et al., 2017) | Phrase-level Edit | F$_\beta$ | ● |
| CLEME (Ours) | Chunk-level Edit | F$_\beta$ | ● |

Table 6: A comparison of mainstream reference-based GEC metrics. GLEU is indeterministic since it involves sampling operation.

experiments.

## 6 Related Work

### 6.1 Reference-based Metrics

Reference-based metrics score GEC systems under the guidance of manually written references. M$^2$ scorer (Dahlmeier and Ng, 2012) determines an optimal edit sequence between a source sentence and a system hypothesis that achieves the highest overlap with the gold-standard annotation. The performance of each system is then represented using the F$_{0.5}$ score. However, optimality in terms of overlap does not guarantee optimality in GEC evaluation. Bryant et al. (2017) showed that M$^2$ scorer exploits its dynamic edit boundary prediction to artificially maximize true positives and minimize false positives, thus producing slightly inflated scores. Therefore, (Bryant et al., 2017) proposed ERRANT, which improves edit extraction using a linguistically-enhanced alignment algorithm and merging rules, improving the alignment of tokens with similar linguistic properties. Despite its effectiveness, ERRANT is language-dependent and bias still exists in multi-reference evaluation. Inspired by BLEU (Papineni et al., 2002) in NMT, Napoles

et al. (2015) proposed GLEU, an n-gram based metric for GEC evaluation. To remedy the shortcoming that $F_{0.5}$ is unable to differentiate a do-nothing system and a bad system unless $TP > 0$, I-measure (Felice and Briscoe, 2015) generates an exact (global optimal) alignment using a three-way alignment algorithm and computes weighted accuracy to score GEC systems in terms of relative textual improvement. The comparison of reference-based GEC metrics is shown in Table 6.

## 6.2 Reference-less Metrics

To avoid the prerequisite of references for GEC evaluation, recent works focus on scoring GEC systems without human-annotated references. Inspired by quality estimation in neural machine translation, Napoles et al. (2016) propose three Grammaticality-Based Metrics (GBMs), which are calculated by a benchmark GEC system or a pretrained ridge regression model. Asano et al. (2017) extend GBMs by introducing three assessment criteria for Grammaticality, Fluency and Meaning preservation (GFM). SOME (Yoshimura et al., 2020) further improves GFM by optimizing each Grammaticality, Fluency and Meaning preservation metric to more closely correlate with human judgements. Scribendi Score (Islam and Magnani, 2021) overcomes the limitations of SOME, requiring neither a benchmark GEC system nor fine-tuning. IMPARA (Maeda et al., 2022) comprises a quality estimator (QE) and similarity estimator (SE) based on BERT (Devlin et al., 2019), which evaluate the quality of GEC output and semantic similarity of two sentences, respectively.

Although recent reference-less metrics may be highly consistent with human judgments, they always suffer from the lack of interpretability and robustness (Bryant et al., 2022), which are crucial factors for GEC evaluation. Additionally, evaluating GEC systems by leveraging pre-trained or fine-tuned language models could pose potential risks. Efficiency of reference-less metrics that rely on language models is also critical if they are used for GEC benchmarks.

## 6.3 Meta Evaluation Methods

It is intractable to determine the best GEC metric. A reasonable GEC metric should take into account multiple factors, including correlation with human judgments, interpretability and efficiency. Inspired by WMT human evaluation campaigns (Callison-Burch et al., 2008), 13 system outputs (including the unchanged source) from the CoNLL-2014 shared task (Ng et al., 2014) were ranked based on human rankings collected by two ranking methods: Expected Wins (EW) and TrueSkill (TS) (Grund-kiewicz et al., 2015). Sakaguchi et al. (2016) collected 8 ($2 \times 2 \times 2$) annotations (minimal and fluency, expert and non-expert, with two corrections each), revealing that GEC metrics work differently across reference sets. Napoles et al. (2019) explored how GEC metrics operate in new domains (formal and informal writing by native English speakers). They constructed a multi-reference GEC test set called GMEG-Data and a new ensemble metric GMEG-Metric.

## 7 Conclusion

This paper proposes CLEME, a reference-based GEC metric that aim to provide unbiased $F_{0.5}$ scores for multi-reference evaluation. We explore evaluating GEC systems based on either the correction dependence assumption or the correction independence assumption. Several possible approaches can be suggested to further improve CLEME. For example, developing (1) a GEC metric that adaptively combines dependent and independent assumptions, and (2) a weighting strategy by utilizing the knowledge of pre-trained model. In the future, we would like to develop CLEME for all languages and admonstrate the effectiveness of CLEME on other languages. It is also worthwhile to explore accuracy-based metrics.

## Limitations

Although CLEME can be extended to other languages, we have not tested its effectiveness in any language other than English. Furthermore, all the reference sets used in our experiments are based on the CoNLL-2014 shared task, a second-language dataset. To demonstrate the robustness of our approaches, further experiments on evaluation datasets with multiple text domains are required. We believe that introducing the correction independence assumption perspective into GEC datasets of other languages and domains could lead to more in-depth analysis and exploration.

While recent PLM-based metrics have shown superior correlations compared to reference-based metrics, including ours on some reference sets, our approach enables the evaluation of GEC systems in an interpretable manner, which is a significant advantage over reference-less metrics. We leave the

exploration of incorporating the PLM's knowledge into CLEME for future work.

## Ethics Statement

In this paper, we verify the effectiveness of our proposed approach using CoNLL-2014, BN-10GEC, and SN-8GEC reference sets, all of which are from publicly available datasets or resources on legitimate websites without sensitive data involved. All the baselines used in our experiments are also publicly available GEC metrics and we have cited the corresponding authors. We confirm that all datasets and baselines used in our experiments are consistent with their intended use.

Additionally, we conduct human evaluation experiments to show the rationality of correction independence assumption. To do so, three postgraduate students specializing in foreign linguistics and applied linguistics were employed as part-time annotators. Each annotator could complete the entire annotation process within approximately 6 working hours. All annotators were paid for their work, with an average salary of approximately $5 per hour.

## Acknowledgements

This research is supported by National Natural Science Foundation of China (Grant No.62276154), Research Center for Computer Network (Shenzhen) Ministry of Education, the Natural Science Foundation of Guangdong Province (Grant No. 2023A1515012914), Basic Research Fund of Shenzhen City (Grant No. JCYJ20210324120012033 and JSGG20210802154402007), the Major Key Project of PCL for Experiments and Applications (PCL2021A06), and Overseas Cooperation Research Fund of Tsinghua Shenzhen International Graduate School (HW2021008).

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

## A  Statistics of Reference Sets

Table 7 presents the statistics of all reference sets involved in our experiments, including In-Corrected-Chunk (ICC) ratio, Unchanged-Chunk (IUC) ratio and Cross-Chunk (CC) ratio. It is worth noting that all reference sets exhibit a low CC ratio with varying ICC and IUC ratios, indicating the rationality and feasibility of evaluating GEC systems following the correction independence assumption.

## B  Hyperparameters

The hyperparameters of our proposed CLEME consist of scale factors $\alpha$ and thresholds. We tune the hyperparameters on CoNLL-2014 and keep them on the other reference sets to demonstrate the adaptability of our method. The hyperparameters of CLEME are listed in Table 8.

## C  Evaluate by Accuracy

Conventional reference-based metrics such as Max-Match ($M^2$) and ERRANT are unable to calculate accuracy because they do not define True Negatives (TNs) [6]. In order to implement the computation of accuracy, CLEME defines TNs as hypothesis unchanged chunks that match the chunks of references. Similar to $F_{0.5}$, accuracy can be computed based on correction dependence or independence assumptions in both corpus- and sentence-level, resulting in four new variants: 1) **CLEME-dependent-acc**, 2) **CLEME-independent-acc**, 3) **SentCLEME-dependent-acc**, and 4) **SentCLEME-independent-acc**.

---

[6]An exception is I-measure (Felice and Briscoe, 2015), which adopts an extended version of the Writer-Annotation-System evaluation scheme (Chodorow et al., 2012).

| Item | CoNLL-2014 | BN-10GEC | E-Minimal | E-Fluency | NE-Minimal | NE-Fluency |
|---|---|---|---|---|---|---|
| # Sents (Length) | 1,312 (23.0) | 1,312 (23.0) | 1,312 (23.0) | 1,312 (23.0) | 1,312 (23.0) | 1,312 (23.0) |
| # Refs (Length) | 2,624 (22.8) | 13,120 (22.9) | 2,624 (23.2) | 2,624 (22.8) | 2,624 (23.0) | 2,624 (22.2) |
| # Edits (Length) | 5,937 (1.0) | 36,677 (1.0) | 4,500 (1.0) | 8,373 (1.1) | 4,964 (0.9) | 11,033 (1.2) |
| # Unchanged Chunks (Length) | 11,174 (4.8) | 93,496 (2.5) | 8,887 (6.3) | 12,823 (3.8) | 10,748 (5.1) | 14,086 (2.9) |
| # Corrected/Dummy Chunks (Length) | 4,994 (1.3) | 26,948 (2.4) | 3,963 (1.2) | 6,305 (1.7) | 4,221 (1.2) | 6,892 (2.6) |
| ICC (Number) | 51.05% (3,031) | 90.66% (33,251) | 62.47% (2,811) | 52.69% (4,412) | 43.84% (2,176) | 43.32% (4,780) |
| IUC (Number) | 45.61% (2,708) | 7.74% (2,837) | 36.6% (1,647) | 42.51% (3,559) | 54.3% (2,698) | 48.92% (5,397) |
| CC (Number) | 3.34% (198) | 1.61% (589) | 0.93% (42) | 4.8% (402) | 1.81% (90) | 7.76% (856) |

Table 7: Statistics of CoNLL-2014 (Ng et al., 2014), BN-10GEC (Bryant and Ng, 2015) and SN-8GEC (Sakaguchi et al., 2016) reference sets. We use ERRANT (Bryant et al., 2017) for edit extraction.

| Configuration | CLEME-dependent(-acc) | CLEME-independent(-acc) | SentCLEME-dependent(-acc) | SentCLEME-independent(-acc) |
|---|---|---|---|---|
| Scale factor of TPs $\alpha_1$ | 2 | 2 | 10 | 10 |
| Scale factor of FPs $\alpha_2$ | 2 | 2 | 10 | 10 |
| Scale factor of FNs $\alpha_3$ | 2 | 2 | - | - |
| Scale factor of TNs $\alpha_4$ | - | - | - | - |
| Threshold of TPs | (0.75, 1.25) | (0.75, 1.25) | (1.00, 10.00) | (2.50, 10.00) |
| Threshold of FPs | (0.75, 1.25) | (0.75, 1.25) | (0.25, 10.00) | (0.25, 1.00) |
| Threshold of FNs | (0.75, 1.25) | (0.75, 1.25) | (1.00, 1.00) | (1.00, 1.00) |
| Threshold of TNs | (1.00, 1.00) | (1.00, 1.00) | (1.00, 1.00) | (1.00, 1.00) |

Table 8: Hyperparameter values used in our experiments.

The results of human correlations are reported in Table 9. Accuracy-based metrics perform very differently at the corpus- and sentence-level, which is similar to the findings (Napoles et al., 2016, 2019). Surprisingly, two accuracy-based corpus-level metrics, i.e., CLEME-dependent-acc and CLEME-independent-acc, result in negative correlations on all reference sets. However, their sentence-level variants, i.e., SentCLEME-dependent-acc and SentCLEME-independent-acc, perform well and achieve the highest correlations on some reference sets. Regarding the disparity between the performance of accuracy-based metrics and $F_{0.5}$-based metrics at the sentence level, one notable difference is their stability or robustness on reference sets with varying numbers of references and annotation styles. $F_{0.5}$-based metrics are more robust to different reference sets, where SentCLEME-(in)dependent achieves comparable correlations with the best metric on all reference sets. However, the performance of accuracy-based metrics lags far behind other metrics on some reference sets (BN-10GEC, E-Minimal and NE-Fluency). A deeper investigation into this phenomenon is needed to understand the instability of accuracy-based metrics. We leave the exploration and further analysis of accuracy-based metric for future work.

## D Detailed Analysis

Table 10 reports the detailed evaluation results of 13 systems on CoNLL-2014. The reason behind the lower TP and FP counts of CLEME as compared to ERRANT is attributed to the chunk partition process, where overlapping edits are merged into chunks. It is worth noting that the FN counts of CLEME are significantly lower than those of ERRANT because of their distinct definition. While ERRANT considers FNs as the edits of references that are not identical to hypotheses, CLEME defines them as the corrected/dummy chunks of references that do not exactly match the chunks of hypotheses. We believe that the definition of ERRANT could be problematic, as it has a tendency to overestimate the FN counts of GEC systems. This may result in an underestimated Recall rate in turn.

**-dependent v.s. -independent.** Comparing the Precision and Recall of (Sent)CLEME-independent to those of (Sent)CLEME-dependent, it is observed that the former has a slightly higher value. This is because (Sent)CLEME-independent has the potential to overestimate the performance of GEC systems, whereas (Sent)CLEME-dependent could result in underestimating the same. It is noteworthy that both metrics provide an upper bound and lower bound for GEC performance, respectively.

**Corpus-level v.s. Sentence-level.** The precision, recall, and $F_{0.5}$ scores of sentence-level metrics are considerably higher than those of corpus-level variants. There might be several factors contributing to this difference, but one possible explanation is that precision and recall values get affected by a limited number of challenging samples that contain numerous corrected/dummy chunks.

Table 9:

| Metric | | CoNLL-2014 EW | CoNLL-2014 TS | BN-10GEC EW | BN-10GEC TS | E-Minimal EW | E-Minimal TS | E-Fluency EW | E-Fluency TS | NE-Minimal EW | NE-Minimal TS | NE-Fluency EW | NE-Fluency TS |
|---|---|---|---|---|---|---|---|---|---|---|---|---|---|
| $M^2$ | $\gamma$ | 0.623 | 0.672 | 0.547 | 0.610 | **0.597** | **0.650** | 0.590 | 0.659 | 0.575 | 0.634 | 0.582 | 0.649 |
| | $\rho$ | 0.687 | 0.720 | 0.648 | 0.692 | 0.654 | 0.703 | 0.654 | 0.709 | 0.577 | 0.648 | 0.648 | 0.703 |
| GLEU | $\gamma$ | **0.701** | **0.750** | **0.678** | **0.761** | 0.533 | 0.513 | **0.693** | **0.771** | -0.044 | -0.113 | **0.674** | **0.767** |
| | $\rho$ | 0.467 | 0.555 | **0.754** | **0.806** | 0.577 | 0.511 | 0.710 | 0.757 | -0.005 | -0.055 | 0.725 | **0.819** |
| ERRANT | $\gamma$ | 0.642 | 0.688 | 0.586 | 0.644 | 0.578 | 0.631 | 0.594 | 0.663 | 0.585 | 0.637 | 0.597 | 0.659 |
| | $\rho$ | 0.659 | 0.698 | 0.637 | 0.698 | 0.742 | 0.786 | 0.720 | 0.775 | 0.747 | 0.797 | **0.753** | 0.797 |
| CLEME-dependent (Ours) | $\gamma$ | 0.648 | 0.691 | 0.602 | 0.656 | 0.594 | 0.644 | 0.589 | 0.654 | 0.595 | 0.643 | 0.612 | 0.673 |
| | $\rho$ | **0.709** | **0.742** | 0.692 | 0.747 | **0.797** | **0.813** | 0.714 | 0.775 | 0.786 | 0.835 | 0.720 | 0.791 |
| CLEME-independent (Ours) | $\gamma$ | 0.649 | 0.691 | 0.609 | 0.659 | 0.593 | 0.643 | 0.587 | 0.653 | **0.601** | **0.647** | 0.611 | 0.672 |
| | $\rho$ | **0.709** | 0.731 | 0.692 | 0.747 | 0.791 | 0.802 | **0.731** | **0.791** | **0.797** | **0.841** | 0.714 | 0.786 |
| CLEME-dependent-acc (Ours) | $\gamma$ | -0.261 | -0.342 | -0.288 | -0.371 | -0.222 | -0.313 | -0.216 | -0.302 | -0.370 | -0.453 | -0.430 | -0.513 |
| | $\rho$ | -0.407 | -0.478 | -0.445 | -0.516 | -0.335 | -0.423 | -0.347 | -0.437 | -0.429 | -0.516 | -0.473 | -0.555 |
| CLEME-independent-acc (Ours) | $\gamma$ | -0.175 | -0.262 | -0.206 | -0.284 | -0.195 | -0.283 | -0.105 | -0.189 | -0.335 | -0.420 | -0.328 | -0.412 |
| | $\rho$ | -0.176 | -0.264 | -0.341 | -0.418 | -0.291 | -0.379 | -0.132 | -0.231 | -0.429 | -0.516 | -0.451 | -0.522 |
| SentM$^2$ | $\gamma$ | 0.871 | 0.864 | 0.567 | 0.646 | 0.805♣ | 0.836♣ | 0.655 | 0.732 | 0.729♣ | 0.785♣ | 0.621 | 0.699 |
| | $\rho$ | 0.731 | 0.758 | 0.593 | 0.648 | 0.806♣ | 0.845♣ | 0.731 | 0.764 | 0.797♣ | 0.846♣ | 0.632 | 0.687 |
| SentGLEU | $\gamma$ | 0.784 | 0.828 | 0.756 | 0.826 | 0.742♣ | 0.773♣ | 0.785 | 0.846 | 0.723♣ | 0.762♣ | 0.778 | 0.848 |
| | $\rho$ | 0.720 | 0.775 | 0.769 | 0.824 | 0.764♣ | 0.797♣ | 0.791 | 0.846 | 0.764♣ | 0.830♣ | 0.768 | 0.846 |
| SentERRANT | $\gamma$ | 0.870 | 0.846 | 0.885 | 0.896 | 0.768♣ | 0.803♣ | 0.806 | 0.732 | 0.710♣ | 0.765♣ | 0.793 | 0.847 |
| | $\rho$ | 0.742 | 0.747 | 0.786 | 0.830 | 0.775♣ | 0.819♣ | 0.813 | 0.764 | 0.780♣ | 0.841♣ | **0.830** | **0.857** |
| SentCLEME-dependent (Ours) | $\gamma$ | 0.876 | 0.844 | **0.915** | **0.913** | 0.806♣ | 0.838♣ | **0.849** | 0.886 | 0.742♣ | 0.795♣ | **0.876** | **0.921** |
| | $\rho$ | 0.824 | 0.808 | 0.835 | 0.874 | 0.775♣ | 0.819♣ | 0.824 | 0.863 | 0.797♣ | 0.846♣ | 0.791 | 0.846 |
| SentCLEME-independent (Ours) | $\gamma$ | 0.868 | 0.857 | 0.855♣ | 0.876♣ | 0.821♣ | 0.856♣ | 0.841 | 0.877 | 0.782♣ | 0.831♣ | 0.852 | 0.896 |
| | $\rho$ | 0.725 | 0.758 | 0.659♣ | 0.714♣ | 0.775♣ | 0.819♣ | 0.808 | 0.846 | 0.819♣ | 0.874♣ | 0.762 | 0.825 |
| SentCLEME-dependent-acc (Ours) | $\gamma$ | 0.828 | 0.857 | 0.650 | 0.719 | 0.808♣ | 0.838♣ | 0.679 | 0.740 | 0.757♣ | 0.811♣ | 0.557 | 0.641 |
| | $\rho$ | 0.813 | 0.841 | 0.682 | 0.740 | 0.830♣ | 0.852♣ | 0.731 | 0.786 | **0.853**♣ | **0.894**♣ | 0.655 | 0.702 |
| SentCLEME-independent-acc (Ours) | $\gamma$ | **0.900**♣ | **0.920**♣ | 0.604♣ | 0.555♣ | 0.693♣ | 0.637♣ | 0.840♣ | **0.891**♣ | 0.791♣ | 0.763♣ | 0.756♣ | 0.822♣ |
| | $\rho$ | **0.830**♣ | **0.849**♣ | 0.363♣ | 0.303♣ | 0.588♣ | 0.544♣ | **0.857**♣ | **0.890**♣ | 0.654♣ | 0.626♣ | 0.747♣ | 0.819♣ |

Table 9: Overview of correlations between reference-based metrics and human judgments. We highlight the **highest** score in bold and the second-highest score with underlines. ♣ We remove unchanged reference sentences for higher correlations due to low-quality annotations. Otherwise, negative correlations are possible.

| Metric | | AMU | CAMB | CUUI | IITB | INPUT | IPN | NTHU | PKU | POST | RAC | SJTU | UFC | UMC |
|---|---|---|---|---|---|---|---|---|---|---|---|---|---|---|
| ERRANT | TP | 483 | 725 | 607 | 28 | 0 | 52 | 409 | 294 | 508 | 319 | 104 | 36 | 311 |
| | FP | 795 | 1329 | 985 | 65 | 0 | 488 | 991 | 697 | 1152 | 794 | 261 | 14 | 774 |
| | FN | 1934 | 1886 | 1946 | 2064 | 2070 | 2078 | 1976 | 2007 | 1985 | 2044 | 2036 | 2069 | 2020 |
| | P | 37.79 | 35.30 | 38.13 | 30.11 | 100.0 | 9.63 | 29.21 | 29.67 | 30.60 | 28.66 | 28.49 | 72.00 | 28.66 |
| | R | 19.98 | 27.77 | 23.78 | 1.34 | 0.00 | 2.44 | 17.15 | 12.78 | 20.38 | 13.50 | 4.86 | 1.71 | 13.34 |
| | $F_{0.5}$ | 32.08 | 33.48 | 34.02 | 5.68 | 0.00 | 6.06 | 25.61 | 23.46 | 27.81 | 23.40 | 14.44 | 7.81 | 23.31 |
| CLEME-dependent | TP | 314 | 482 | 379 | 17 | 0 | 33 | 266 | 195 | 333 | 203 | 69 | 24 | 213 |
| | w/o LW | 382 | 588 | 471 | 22 | 0 | 39 | 330 | 246 | 412 | 254 | 85 | 32 | 216 |
| | FP | 872 | 1392 | 1034 | 72 | 0 | 529 | 975 | 776 | 1246 | 782 | 292 | 19 | 844 |
| | w/o LW | 815 | 1303 | 964 | 67 | 0 | 488 | 905 | 709 | 1144 | 782 | 272 | 18 | 788 |
| | FN | 1182 | 987 | 1169 | 1564 | 1592 | 1445 | 1191 | 1259 | 1158 | 1278 | 1471 | 1583 | 1266 |
| | w/o LW | 1345 | 1132 | 1333 | 1751 | 1782 | 1634 | 1366 | 1426 | 1333 | 1453 | 1657 | 1772 | 1439 |
| | TN | 6312 | 6347 | 6245 | 6313 | 6308 | 6412 | 6295 | 6449 | 6310 | 6324 | 6280 | 6377 | |
| | P | 26.45 | 25.74 | 26.81 | 19.29 | 100.0 | 5.85 | 21.42 | 20.06 | 21.07 | 20.60 | 19.02 | 56.40 | 20.14 |
| | R | 20.97 | 32.84 | 24.48 | 1.09 | 0.00 | 2.22 | 18.23 | 13.39 | 22.31 | 13.71 | 4.45 | 1.52 | 14.40 |
| | $F_{0.5}$ | 25.14 | 26.90 | 26.31 | 4.45 | 0.00 | 4.41 | 20.69 | 18.24 | 21.31 | 18.72 | 11.50 | 6.85 | 18.65 |
| SentCLEME-dependent | P | 63.05 | 41.07 | 57.60 | 95.27 | 100.0 | 67.45 | 55.17 | 63.62 | 53.26 | 65.05 | 82.90 | 98.70 | 61.78 |
| | R | 48.87 | 59.94 | 51.21 | 32.37 | 31.33 | 36.07 | 49.71 | 44.84 | 50.61 | 44.57 | 36.06 | 32.15 | 44.43 |
| | $F_{0.5}$ | 36.24 | 32.94 | 37.39 | 31.51 | 31.33 | 23.25 | 32.24 | 32.56 | 33.34 | 32.37 | 31.93 | 32.30 | 31.46 |
| CLEME-independent | TP | 318 | 488 | 392 | 17 | 0 | 33 | 272 | 196 | 339 | 204 | 69 | 24 | 214 |
| | w/o LW | 388 | 596 | 487 | 22 | 0 | 39 | 338 | 248 | 420 | 255 | 85 | 32 | 262 |
| | FP | 864 | 1382 | 1016 | 72 | 0 | 529 | 965 | 773 | 1236 | 781 | 292 | 19 | 843 |
| | w/o LW | 809 | 1295 | 948 | 67 | 0 | 488 | 897 | 707 | 1136 | 781 | 272 | 18 | 787 |
| | FN | 928 | 701 | 884 | 1362 | 1393 | 1246 | 937 | 1022 | 883 | 1025 | 1266 | 1372 | 1026 |
| | w/o LW | 1029 | 778 | 984 | 1497 | 1530 | 1382 | 1045 | 1129 | 990 | 1135 | 1398 | 1506 | 1136 |
| | TN | 6629 | 6701 | 6597 | 6567 | 6560 | 6664 | 6617 | 6611 | 6793 | 6628 | 6583 | 6546 | 6680 |
| | P | 26.90 | 26.11 | 27.85 | 19.29 | 100.0 | 5.85 | 22.00 | 20.23 | 21.50 | 20.69 | 19.02 | 56.40 | 20.22 |
| | R | 25.53 | 41.06 | 30.71 | 1.25 | 0.00 | 2.57 | 22.52 | 16.10 | 27.72 | 16.59 | 5.14 | 1.75 | 17.23 |
| | $F_{0.5}$ | 26.61 | 28.16 | 28.38 | 4.97 | 0.00 | 4.66 | 22.10 | 19.24 | 22.51 | 19.71 | 12.35 | 7.77 | 19.54 |
| SentCLEME-independent | P | 65.36 | 45.39 | 60.92 | 95.27 | 100.0 | 67.51 | 57.44 | 65.03 | 56.26 | 66.65 | 83.17 | 98.70 | 63.36 |
| | R | 57.20 | 70.15 | 60.76 | 36.87 | 35.29 | 40.56 | 58.08 | 52.20 | 59.83 | 52.14 | 41.31 | 36.59 | 51.94 |
| | $F_{0.5}$ | 42.00 | 39.63 | 43.76 | 35.49 | 35.29 | 25.90 | 38.13 | 37.43 | 39.38 | 37.04 | 36.25 | 36.48 | 36.59 |

Table 10: Detailed evaluation results across 13 GEC systems on CoNLL-2014.

|          | Chunk 1 | Chunk 2 | Chunk 3 | Chunk 4 | Chunk 5 | Chunk 6 | Chunk 7 |
|----------|---------|---------|---------|---------|---------|---------|---------|
| Source   | If      |         | not     | their family then | who else | that are | willing to do that ? |
| Ref. 1   | If      |         | not     | their family then | who else | will be | willing to do that ? |
| Ref. 2   | If      |         | not     | their family then | who else | would be | willing to do that ? |
| Ref. 3   | If      |         | not     | from your family then | who else | is | willing to do that ? |
| Ref. 4   | If      |         | not     | their family , then | who else | will be | willing to do that ? |
| Ref. 5   | If      |         | not     | their family then | who else | will be | willing to do that ? |
| Ref. 6   | If      |         | not     | their family | who else | would be | willing to do that ? |
| Ref. 7   | If      |         | not     | their family then | who else | will be | willing to do that ? |
| Ref. 8   | If      |         | not     | their family , | who else | is | willing to do that ? |
| Ref. 9   | If      | family do | not   | help then | who else | would be | willing to do that ? |
| Ref. 10  | If      |         | not     | their family , then | who else | is | willing to do that ? |

Table 11: A case of correction independence. We apply chunk partition to the source and all the references.

|          | Chunk 1 | Chunk 2 | Chunk 3 | Chunk 4 |
|----------|---------|---------|---------|---------|
| Source   | For not | use     | car     | .       |
| Ref. 1   | Not for | use     | with a car | .    |
| Ref. 2   | Do not  | use     | in the car | .    |
| Ref. 3   | Car not for | use |         | .       |
| Ref. 4   | Can not | use     | the car | .       |

|          | Chunk 1 | Chunk 2 | Chunk 3 | Chunk 4 | Chunk 5 | Chunk 6 |
|----------|---------|---------|---------|---------|---------|---------|
| Source   | One person if do n't | have good health | that means | so many things | they could lost | . |
| Ref. 1   | If a person does n't | have good health | , | so many things | could be lost | . |
| Ref. 2   | If one person does not | have good health | , that means they could lose | so many things | | . |
| Ref. 3   | If one person does n't | have good health | , that means they could lose | so many things | | . |
| Ref. 4   | If one person does n't | have good health | , that means | so many things | they could lost | . |

|          | Chunk 1 | Chunk 2 | Chunk 3 | Chunk 4 | Chunk 5 | Chunk 6 |
|----------|---------|---------|---------|---------|---------|---------|
| Source   | 今天    | 听天气预报说 | 今     | 天      | 还有天气 | 冷。    |
| Ref. 1   |         | 听天气预报说 | 今     | 天      | 天气    | 冷。    |
| Ref. 2   | 今天    | 听天气预报说 |        | 天      | 气还会变 | 冷。    |
| Ref. 3   |         | 听天气预报说 | 今     | 天      | 天气还会变 | 冷。  |

|          | Chunk 1 | Chunk 2 | Chunk 3 | Chunk 4 | Chunk 5 | Chunk 6 | Chunk 7 | Chunk 8 |
|----------|---------|---------|---------|---------|---------|---------|---------|---------|
| Source   | 所以    | 我从小到现在在这些快餐 |        | 吃饭的机会很少 | 。 | 对我来说每 | 次 | 饭都很重要。 |
| Ref. 1   | 所以    | 我从小到现在在这些快餐 | 店     | 吃饭的机会很少 | 。 | 对我来说每 | 顿 | 饭都很重要。 |
| Ref. 2   | 所以    | 我从小到现在在这些快餐 | 店     | 吃饭的机会很少 | 。 | 对我来说每 | 次吃 | 饭都很重要。 |
| Ref. 3   |         | 我从小到现在在这些快餐 | 店     | 吃饭的机会很少 | ，所以 | 对我来说每 | 次吃 | 饭都很重要。 |
| Ref. 4   |         | 我从小到现在在这些快餐 | 店     | 吃饭的机会很少 | ，所以 | 对我来说每 | 顿 | 饭都很重要。 |

Table 12: More cases of chunk partition. These tables are automatically generated by CLEME. The first two cases are from JELEG-*dev*, and the next two cases are from MuCGEC-*dev*.

## E  Correction Independence

We introduce the term *correction independence* to describe a pair of chunks where the correction of each chunk is not related to the correction of the other, as illustrated in Table 11. Specifically, chunk 2 and chunk 4 are considered correction-dependent because the correction of chunk 2 *family do* from Ref.9 must be matched with the correction of chunk 4 *help then* from Ref.9. However, chunk 6 is correction-independent with chunk 2 (or 4) since the correction of chunk 6 has no impact on the correction of chunk 2 (or 4).

## F  More cases

We list more cases in Table 12, which involve JFLEG (Napoles et al., 2017) for English, and MuCGEC (Zhang et al., 2022) for Chinese.