# OpenReview forum: "CLEME: Debiasing Multi-reference Evaluation for Grammatical Error Correction"
_EMNLP/2023/Conference — EMNLP 2023 Main_

### Official Review · Reviewer_oXwi · 2023-07-25

**Typos Grammar Style And Presentation Improvements:** + line 76
**Soundness:** 4

**Excitement:**

4: Strong: This paper deepens the understanding of some phenomenon or lowers the barriers to an existing research direction.

**Missing References:**

+ MuCGEC (NAACL'22), a Chinese multi-reference dataset along with a span-level metric proposed.

**Paper Topic And Main Contributions:**

This paper introduces the Chunk-Level Multi-Reference Metric (CLEME) to address a significant issue in Grammatical Error Correction (GEC): multi-reference evaluation. By comparing all references and the source input, CLEME converts them into chunk sequences with consistent boundaries. The authors assert that CLEME can eliminate bias in multi-reference GEC evaluation, as the number of true positives (TP) remains equal for all references.

Additionally, the authors assume that all corrections in the references are independent, which is referred to as the correction independent assumption. This allows them to calculate metrics by comparing hypothesis corrections with all golden corrections across references.

The final experimental results demonstrate the effectiveness of the proposed CLEME metric in addressing the targeted issue.

**Questions For The Authors:**

+ Can you provide more detailed information about the human evaluation?
+ Can you provide more details about the CLEME toolkit, such as the core pseudo code, the screenshot of the visualization results, etc.

**Reasons To Accept:**

+ The paper is well-written and easy to understand.
+ This paper highlights an important yet under-explored aspect of GEC, specifically, reducing bias in multi-reference evaluation. The authors provide ample preliminary experiments to support their motivations, such as the chunk-based evaluation and the independent correction assumption.
+ The final metric designed in this study demonstrates strong correlations with human judgments, outperforming most previous metrics, including both reference-based and reference-free approaches.



**Reasons To Reject:**

+ In my opinion, the chunk-based design has a potentially unreasonable aspect. Using the example in Figure 1 as an illustration, if a GEC system correctly deletes *the*, it would receive a true positive (TP) when evaluated using ERRANT, while it would receive a false positive (FP) when evaluated using CLEME. This example suggests that CLEME might be somewhat too strict due to its chunk-based evaluation setting, which could lead to an underestimation of GEC systems' performance.
+ I suggest the authors conduct more experiments on other languages such as Chinese, which already has some multi-reference datasets, e.g., YACLC and MuCGEC, to further confirm the superiority of their metric.

**Reproducibility:**

5: Could easily reproduce the results.

**Reviewer Confidence:**

4: Quite sure. I tried to check the important points carefully. It's unlikely, though conceivable, that I missed something that should affect my ratings.

---

> ### Author Rebuttal · Authors · 2023-08-28
>
> Thanks for your appreciation of our work's contributions. We are honored to have the opportunity to discuss with you.
>
> ---
>
> **Q1**: Is CLEME somewhat too strict to count a true positive (TP), which could lead to an underestimation of GEC performance?
>
> **A1**: We acknowledge that CLEME may be too strict in recognizing a TP if a chunk is too long. To alleviate this problem, we have introduced the **length weighting** to encourage the matching of longer chunks.
>
> ---
>
> **Q2**: Can more experiments on other languages be conducted?
>
> **A2**: Unfortunately, only CoNLL-2014 datasets are equipped with human ranking, which is crucial for quantitative experiments. However, we will present **qualitative results and analyses** on other multi-reference datasets, including BEA-19 [1], JFLEG [2] and MuCGEC [3] in our final version. **We will also cite relevant papers in our final version**.
>
> ---
>
> **Q3**: Can you provide more detailed information about the human evaluation?
>
> **A3**: Yes, we will provide the details about the human evaluation in Section 4.
>
> ---
>
> **Q4**: Can you provide more details about the CLEME toolkit, such as the core pseudo code, the screenshot of the visualization results, etc.
>
> **A4**: Absolutely, we will include more information about CLEME  in the final version. We also plan to **open-source a comprehensive GEC evaluation toolkit**, including CLEME, ERRANT, MaxMatch, GLEU, etc.
>
> ---
>
> **References**
>
> [1] The BEA-2019 Shared Task on Grammatical Error Correction. ACL Workshop 2019.
>
> [2] JFLEG: A Fluency Corpus and Benchmark for Grammatical Error Correction. EACL 2017.
>
> [3] MuCGEC: a Multi-Reference Multi-Source Evaluation Dataset for Chinese Grammatical Error Correction. NAACL 2022.

---

### Official Review · Reviewer_WBcg · 2023-08-05

**Soundness:** 4

**Excitement:**

4: Strong: This paper deepens the understanding of some phenomenon or lowers the barriers to an existing research direction.

**Missing References:**

- [JFLEG: A Fluency Corpus and Benchmark for Grammatical Error Correction](https://aclanthology.org/E17-2037) (Napoles et al., EACL 2017)

**Paper Topic And Main Contributions:**

This paper proposes a new reference-based evaluation metric for grammatical error correction (GEC), CLEME, which is characterized by the consideration of multiple reference sentences and the calculation of F0.5 at the chunk level. The preliminary study (Section 2) investigates chunk-level boundaries, and the proposed metric is designed based on the analysis. The effectiveness of the metric was evaluated in a comparison experiment with existing reference-based metrics.

**Questions For The Authors:**

Question A: The experimental setup about ♣ in the tables is unclear. In what case is the unchanged reference set excluded?

Question B: What are the advantages of CLEME-independent compared to CLEME-dependent?


**Reasons To Accept:**

- The proposed metric is a simple algorithm and achieves performance comparable to existing methods.
- The proposed metric design is based on preliminary research, and the experiments and analysis are comprehensive.
- The manuscript is well written.


**Reasons To Reject:**

The correction independence assumption is an interesting finding, but it is unclear whether it can be applied to GEC tasks in general. Three datasets are used in this paper, all based on CoNLL-2014 (line 285). CoNLL-2014 is the standard evaluation dataset in GEC, but experiments with other data sets, for example, JFLEG, should also be considered.

**Reproducibility:**

4: Could mostly reproduce the results, but there may be some variation because of sample variance or minor variations in their interpretation of the protocol or method.

**Reviewer Confidence:**

3: Pretty sure, but there's a chance I missed something. Although I have a good feel for this area in general, I did not carefully check the paper's details, e.g., the math, experimental design, or novelty.

**Typos Grammar Style And Presentation Improvements:**

- line 359: Suggests → suggests

---

> ### Author Rebuttal · Authors · 2023-08-28
>
> Thanks for your valuable comments. We attach great importance to your questions and hope to address your concerns.
>
> ---
>
> **Q1**: Can CLEME be applied to other GEC datasets, like JFLEG?
>
> **A1**: Yes, it can. We have not considered experiments on other datasets like JFLEG [1] because of **unavailability of human evaluation results**. However, we are able to provide **qualitative results** on these datasets. Below are some evaluation examples from JELEG. **We will enhance our evaluation analysis on other datasets and cite relevant papers in our final version**.
>
> | Sentence | chunk-0 *   | chunk-1 | chunk-2  *  | chunk-3 |
> | -------- | ----------- | ------- | ---------- | --------- |
> | Source   | For not     | use     | car        | .         |
> | Ref. 0   | Not for     | use     | with a car | .         |
> | Ref. 1   | Do not      | use     | in the car | .         |
> | Ref. 2   | Car not for | use     |            | .         |
> | Ref. 3   | Can not     | use     | the car    | .         |
>
>
> | Sentence | chunk-0 *              | chunk-1          | chunk-2  *                   | chunk-3      | chunk-4 *      | chunk-5 |
> | -------- | ---------------------- | ---------------- | ---------------------------- | -------------- | --------------- | --------- |
> | Source   | One person if do n't   | have good health | that means                   | so many things | they could lost | .         |
> | Ref. 0   | If a person does n't   | have good health | ,                            | so many things | could be lost   | .         |
> | Ref. 1   | If one person does not | have good health | , that means they could lose | so many things |                 | .         |
> | Ref. 2   | If one person does n't | have good health | , that means they could lose | so many things |                 | .         |
> | Ref. 3   | If one person does n't | have good health | , that means                 | so many things | they could lost | .         |
>
>
> Similar to Table 5 in our paper, CLEME can automatically generate visualization of chunk partition results for JFLEG. Since JELEG is built upon fluency correction, we guess CLEME-dependent is more applicable.
>
> ---
>
> **Q2**: The experimental setup about ♣ in the tables is unclear. In what case is the unchanged reference set excluded?
>
> **A2**: We exclude the unchanged reference set if we observe **slight or even negative correlations** for some or most metrics. This intervention is only applied to sentence-level metrics since they are more sensitive to noisy reference sets. For example, the NE-Minimal reference set contains many under-corrections since it was annotated by non-experts following the minimal edit principle. We observe slight or negative correlations for all the metrics in the NE-Minimal reference set without intervention. Similarly, we observe slight correlations in the E-Minimal reference set due to the same reason. An exception is BN-10GEC, where we observe a significant drop for SentPT-M^2 and SentCLEME-independent. Further investigation on the **sensitivity of GEC metrics** is part of our future work.
>
> ---
>
> **Q3**: What are the advantages of CLEME-independent compared to CLEME-dependent?
>
> **A3**: CLEME-independent is built upon the new evaluation assumption, i.e., **correction independence assumption**. It is a **relaxed variant of correction dependence assumption**, and is particularly applicable when grammatical error density is low. It become apparent that texts written by native speakers generally exhibit a lower grammatical error density than L2-speak datasets (CoNLL-2014). In such cases, correction independence assumption seems to be a promising means to **alleviate the scarcity of human-written references** because different grammatical errors within a sentence are more likely to be independent of each other.
>
> ---
>
> **References**
>
> [1] JFLEG: A Fluency Corpus and Benchmark for Grammatical Error Correction. EACL 2017.

---

### Official Review · Reviewer_XNd5 · 2023-08-18

**Typos Grammar Style And Presentation Improvements:** Increase the font size in the figures…
**Soundness:** 4

**Excitement:**

4: Strong: This paper deepens the understanding of some phenomenon or lowers the barriers to an existing research direction.

**Paper Topic And Main Contributions:**

The paper proposes CLEME, a new metric for the evaluation of grammatical error correction (GEC). The novelty is based on two major claims: 1) When there are multiple references, the alignment of the edits (corrections) should be consistent across all references, not just one at a time, as opposed to an established metric called ERRANT. 2) Edits are often independent of each other; therefore corrections can and should take multiple of the references into account at the same time (see Figures 1 and 2 for an illustration). To support the second claim, the authors perform human evaluation (Sec. 4.2).

**Questions For The Authors:**

Could you please clarify, not just in your response to the reviewers, but in the paper itself:

A) Can you elaborate why the IAA you get is to be considered good "given the subjectivity of the task" in Section 4.2? Why is it subjective to tell whether two edits are dependent of each other or not? Is it not mostly about basic grammar?

B) Can you elaborate on the impact of local vs. less local errors (mentioned above)? How would CLEME manage with more challenging error types?

C) Could you explain more clearly what reference-based and reference-less evaluation mean, mentioned for instance in the introduction?

D) You sort of make it sound as if you have come up with the idea of merging adjacent chunks, but in the ERRANT algorithm there are different merge strategies as well: rule-base, all-split, all-merge, all-equal. Could you describe more clearly your contribution in relation to theirs? (It is not that difficult to guess, but it would be good to elaborate.)

**Reasons To Accept:**

The paper is well written and can be followed rather easily. The experiments and analysis are thorough and multifaceted. The results are decent in comparison to a handful of benchmark models on a number of standard data sets.

The problem is relevant. It addresses real challenges observed for existing models. My guess is that multi-reference or multi-hypothesis systems will become more popular with the new opportunities offered by text generation based on large language models (LLMs). The community will need algorithms that take into account multiple parallel references (or hypotheses) and align corresponding spans (chunks) successfully, treating them as dependent or independent of each other, across references.


**Reasons To Reject:**

Table 2 suggests that CLEME performs decently compared to the baselines, but not phenomenally well. The approach that the authors seem to advocate the most (SentCLEME-independent) works best on only 3 out of 12 data sets (however, being second best on 5 additional sets).

The independence claim does get strong support from the three annotators individually (> 90% of the cases match the assumption), but strangely enough the annotators seem to disagree which of the cases are such, because the Cohen's kappa values are not that high (Table 3). However, the authors suggest that the IAA statistics are good considering the subjectivity of the task. But I am not sure why that should be so.

The algorithm proposed by the authors probably works well only when the errors are very local (as mentioned in the introduction section). If there are larger errors involving incorrect word order, my guess is that the approach might "collapse", producing one single correction chunk spanning the entire sentence. (However, other metrics, such as ERRANT, do have the same limitations.) It would be interesting to the community to see algorithms and metrics that can cope with more challenging text with multiple errors.

Figures 1 and 2 (with Figure 1, in particular) is unreadable because of the minimal font used. Since it is crucial to be able to read the text, in order to understand these figures and the algorithm, the font size should be bigger. I have been told that footnote size is the smallest acceptable font size for any text in a paper. Reading a paper in digital format does give the reader the opportunity to zoom in on the figure, but that is not good enough. The paper already occupies the maximum length allotted, so cramming in additional info by using very small text in figures, makes one wonder whether this is one way of extending the actual length of the paper beyond the given limits.

**Reproducibility:**

4: Could mostly reproduce the results, but there may be some variation because of sample variance or minor variations in their interpretation of the protocol or method.

**Reviewer Confidence:**

4: Quite sure. I tried to check the important points carefully. It's unlikely, though conceivable, that I missed something that should affect my ratings.

---

> ### Author Rebuttal · Authors · 2023-08-28
>
> We are grateful for your positive comments and insightful reviews. Please find below our point-by-point responses:
>
> ---
>
> **Q1**: Can you elaborate why the IAA you get is to be considered good "given the subjectivity of the task" in Section 4.2? Why is it subjective to tell whether two edits are dependent of each other or not? Is it not mostly about basic grammar?
>
> **A1**: The IAA of our paper is compared with previous studies. [1] attempted to compute IAA at the sentence level. Three raters were asked simply to decide whether 200 sentences were correct or not. The authors reported **IAA of just 0.16, 0.4 and 0.23**. Therefore, we claim in our paper that the IAA of correction independence is **relatively higher**. On the other hand, our annotators may be confused with some hard cases of correction independence, when two chunks are close. More discussion on IAA of GEC can refer to [2].
>
> ---
>
> **Q2**: Can you elaborate on the impact of local vs. less local errors (mentioned above)? How would CLEME manage with more challenging error types?
>
> **A2**: If there are "global" errors spanning the whole sentence, CLEME, and even other metrics such as ERRANT, will indeed "collapse". However, it is an extreme case, and we have not observed similar situations in CoNLL-2014 [3], BEA-dev [4] (not shown in the paper), JFLEG [5] (not shown in the paper) and Chinese MuCGEC [6] (not shown in the paper). **If collapse happens, the quality of the reference set should be checked first**. It is also feasible to detect latent invalid references leading to collapse using CLEME in future work. Given that a source with multiple references, **CLEME can be improved to filter the references that lead to collapse and use the remaining references to evaluate**.
>
> ---
>
> **Q3**: Could you explain more clearly what reference-based and reference-less evaluation mean, mentioned for instance in the introduction?
>
> **A3**: Both reference-based and reference-less metrics are distinguished based on **whether they rely on human-annotated reference sets**. The latter metrics usually evaluate GEC systems' output using PLMs. However, **it is also possible to combine reference-based metrics and PLMs, which is one of our future work**.
>
> ---
>
> **Q4**: You sort of make it sound as if you have come up with the idea of merging adjacent chunks, but in the ERRANT algorithm there are different merge strategies as well: rule-base, all-split, all-merge, all-equal. Could you describe more clearly your contribution in relation to theirs? (It is not that difficult to guess, but it would be good to elaborate.)
>
> **A4**: This suggestion is highly valuable. All the merge strategies in ERRANT are only available in the single-reference setting. The chunk partition, proposed in our paper, can be treated as **a variant of merge strategies**, which aims to **debias evaluating in the multi-reference setting**. We will elaborate our contribution in relation to theirs in our final version.
>
> ---
>
> **Q5**: Figures 1 and 2 (with Figure 1, in particular) is unreadable because of the minimal font used.
>
> **A5**: We acknowledge that the font used in Figures 1 and 2, particularly in Figure 1, makes them difficult to read. We will ensure that these Figures are redrawn using a more readable font in the final version.
>
> ---
>
> **References**:
>
> [1] Annotating ESL errors: Challenges and rewards. NAACL Workshop 2010.
>
> [2] How Far are We from Fully Automatic High Quality Grammatical Error Correction? ACL 2015.
>
> [3] The CoNLL-2014 Shared Task on Grammatical Error Correction. ACL Shared Task 2014.
>
> [4] The BEA-2019 Shared Task on Grammatical Error Correction. ACL Workshop 2019.
>
> [5] JFLEG: A Fluency Corpus and Benchmark for Grammatical Error Correction. EACL 2017.

---

### Meta-Review · Area_Chair_yb1g · 2023-09-15

**Recommendation:** 4

**Metareview:**

The paper proposes a new metric for the evaluation of grammatical error correction (GEC) that takes advantage of multiple references and operates on chunk level.

The reviewers agree that this is a well-written paper and contains extensive experiments and analyses. The proposed technique is relatively simple but effective. However, the reviewers raised some questions about the proposed methods, in particular the minor improvements compared to existing metrics, the generalizability of the independence assumptions, and the behavior of the proposed method in cases of non-local errors.

---

### Decision · Program_Chairs · 2023-10-07

**Decision:**

Accept-Main

**Comment:**

The paper proposes a new metric for the evaluation of grammatical error correction (GEC) that takes advantage of multiple references and operates on chunk level.

The reviewers agree that this is a well-written paper and contains extensive experiments and analyses. The proposed technique is relatively simple but effective. However, the reviewers raised some questions about the proposed methods, in particular the minor improvements compared to existing metrics, the generalizability of the independence assumptions, and the behavior of the proposed method in cases of non-local errors.